# Diagnostic Strategies for Thyroid Nodules Based on Ultrasonographic Findings in Japan

**DOI:** 10.3390/cancers13184629

**Published:** 2021-09-15

**Authors:** Hiroki Shimura, Yoshiko Matsumoto, Tsukasa Murakami, Nobuhiro Fukunari, Masafumi Kitaoka, Shinichi Suzuki

**Affiliations:** 1Department of Laboratory Medicine, School of Medicine, Fukushima Medical University, Fukushima 960-1295, Japan; 2Department of Thyroid and Endocrinology, School of Medicine, Fukushima Medical University, Fukushima 960-1295, Japan; a0001568@fmu.ac.jp (Y.M.); shsuzuki@fmu.ac.jp (S.S.); 3Department of Endocrinology, Noguchi Thyroid Clinic and Hospital Foundation, Oita 874-0902, Japan; murakami@noguchi-med.or.jp; 4Thyroid Center, Showa University Northern Yokohama Hospital, Kanagawa 224-8503, Japan; fukunari@med.showa-u.ac.jp; 5Department of Endocrinology and Metabolism Center, IMS Miyoshi General Hospital, Saitama 354-0041, Japan; kitaoka.masafumi@ims.gr.jp

**Keywords:** thyroid cancer, thyroid nodule, ultrasonography, fine needle aspiration cytology, guideline

## Abstract

**Simple Summary:**

In recent years, the incidence of thyroid cancer has been increasing worldwide, mainly due to the widespread use of imaging examination methods, such as ultrasonography. In Japan, the risk of overdiagnosis due to the use of diagnostic imaging has been discussed since the 1990s, and measures have been taken to reduce the risk of overdiagnosis and overtreatment by establishing criteria for the implementation of fine needle aspiration cytology and non-surgical follow-up. This article outlines the diagnostic criteria and management guidelines in Japan in comparison with those published in other countries.

**Abstract:**

In recent years, the incidence of thyroid cancer has been increasing worldwide, which is believed to be mainly due to the widespread use of imaging examinations, such as ultrasonography. In this context, ultrasonography has become increasingly important because it can evaluate not only the presence or absence of nodules, but also the detailed characteristics of the nodule, making it possible to diagnose benign or malignant nodules before cytology is performed. In Japan, the third edition of the sonographic diagnostic criteria for thyroid nodules is currently widely used, and its content is similar to that of recent meta-analyses and guidelines from medical societies in other countries. In addition, since overdiagnosis of very-low-risk thyroid cancer has recently become an issue, criteria for the implementation of fine needle aspiration cytology (FNAC) have been published by various countries. The Japan Society of Breast and Thyroid Sonology provides guidelines for FNAC implementation for solid and cystic nodules. In the United States, the ATA, NCCA, and ACR have published guidelines, whereas in Europe, the ESMO and ETA have done the same. All of these guidelines used to classify risk are based on nodule size and sonographic findings. This article outlines the diagnostic criteria and management guidelines in Japan in comparison with those published in other countries. Case studies using actual images were also performed to examine the differences in the FNAC guidelines.

## 1. Introduction

The incidence of thyroid cancer is increasing worldwide, and one of the reasons for this is the increase in the incidental detection of thyroid cancers due to the widespread use of high-performance ultrasound equipment, as well as environmental factors [1]. In particular, it has been reported that the incidence of thyroid cancer increased 15-fold in 2011 compared to 1993 in Korea, where a cancer screening program was started in 1999, due to an increase in the number of women receiving thyroid ultrasound examinations along with breast ultrasound examinations [2]. Furthermore, since the prevalence of thyroid cancer is rising in various countries around the world, this topic has become a point of debate, resulting in the publication of reports and recommendations that warn of overdiagnosis [3].

In Japan, the risk of the overdiagnosis of thyroid cancer in thyroid ultrasonography has been pointed out since the 1990s, and medical follow-up has been conducted for very-low-risk micropapillary carcinomas [4]. In addition, the Japan Association of Breast and Thyroid Sonology (JABTS) proposed criteria for the implementation of fine needle aspiration cytology (FNAC) in *Thyroid Ultrasound—A Guidebook for Diagnosis and Management* (2nd Edition) in 2012 [4]. In particular, FNAC should be performed only for solid nodules 5.1–10.0 mm in diameter that have most of the malignant findings in the ultrasonographic criteria for thyroid nodules published by The Japan Society of Ultrasonics in Medicine (JSUM) [4]. As a result, FNAC should be carefully considered for very-low-risk thyroid cancers that are subject to active surveillance. The appropriate evaluation of ultrasonographic findings of nodules and the widespread use and standardization of FNAC criteria are important issues.

In this article, we review how thyroid nodules are diagnosed in Japan, where they are increasingly detected using diagnostic imaging such as ultrasonography, and compare them with the criteria in other countries.

## 2. Sonographic Features for the Diagnosis and Risk Assessment of Thyroid Nodules

### 2.1. Ultrasonographic Criteria for Thyroid Nodules in Japan

Ultrasonography is recommended for the diagnosis and risk assessment of thyroid nodules detected on the basis of symptoms or palpation, and those detected incidentally through diagnostic imaging. A number of studies have shown that ultrasound findings are important for the differential diagnosis of benign and malignant thyroid nodules, and diagnostic criteria have been published in various countries.

In 2011, the JSUM published ultrasound diagnostic criteria for thyroid nodules (Table 1) [5]. To establish these criteria, the members of the JABTS prepared reference images of benign and malignant nodules, and 17 members evaluated the ultrasound findings of each image on a 3-point scale [6]. As a result, the shape, border character, and internal echo level showed the highest sensitivity and specificity, followed by the edge definition and heterogeneity of internal echoes. These characteristics were selected as the primary findings. Furthermore, multivariate analyses revealed that the former three items were able to discriminate papillary carcinomas in more than 90% of cases, with good sensitivity and specificity [6]. On the other hand, the marginal hypoechoic zone and fine strong echoes were considered secondary findings because the statistical differences between benign and malignant nodules were lower than those in the primary findings. In addition, the present diagnostic criteria focus not only on papillary thyroid carcinoma, but also on other thyroid malignancies, such as follicular carcinoma, medullary carcinoma, malignant lymphoma, and undifferentiated carcinoma [5]. These criteria allow for a wide range of thyroid malignancies by adding the statement that malignant diseases that can present with benign findings include microinvasive follicular carcinoma, micropapillary carcinoma less than 10 mm, medullary carcinoma, and malignant lymphoma.

Recently, Japanese researchers affiliated with JABTS and JSUM conducted a multicenter study to investigate the importance of taller-than-wider signs of thyroid nodules in the diagnosis of malignancy [7]. The results showed that the sensitivity was relatively low (36.5%), but the specificity was high (88.5%). The results also showed that there was a significant difference in the diagnosis of papillary thyroid carcinoma from benign nodules, but no significant difference between follicular carcinoma and benign nodules. It could be anticipated that a taller-than-wide sign will be added, in addition to the irregularity of shape, as a shape evaluation method into the diagnostic criteria of the JSUM.

**Table 1 cancers-13-04629-t001:** Sonographic characteristics potentially suggesting malignancy.

Sonographic Characteristics	JSUM (2011) [5]	Brito (2014) [8]	Campanella (2014) [9]	Remonti (2015) [10]
Shape	Irregular	Taller than wider	Height greater than width	Taller than wide
Border (Definition)	Ill-defined			
(Character)	Jagged	Infiltrative margins	Irregular margins	Irregular margins
Internal echoes (Level)	Hypoechoic	Hypoechoic	Hypoechogenicity	Hypoechogenicity
(Heterogeneity)	Heterogeneous			Heterogeneity
Strong echoes	Multiple and fine	Internal calcification	Microcalcifications	Microcalcifications
Marginal hypoechoic zone (halo)	Irregular or absent		Absent halo sign	Halo absent
Solid/cystic		Solid nodule	Solid nodule structure	Solid
Internal blood flow		Increased blood flow	Intranodular vascularization	Central vascularization
Elasticity				Absence of elasticity
Nodule size			Nodule size ≥ 4 cm	
Single/multiple			Single nodule	Solitary

### 2.2. Meta-Analysis of Ultrasound Findings That Can Distinguish between Benign and Malignant Tumors

Many reports have been published worldwide showing useful ultrasound findings for the diagnosis and risk assessment of benign and malignant thyroid nodules. Recently, meta-analyses of the accumulated evidence have reported useful ultrasound findings that contribute to the diagnosis of benign and malignant thyroid nodules (Table 1) [8,9,10].

In the meta-analysis by Brito et al. [8], the sonographic feature of the nodules that had the highest odds ratio for the diagnosis of malignancy was the taller-than-wider feature (11.14 (95% CI 6.6–18.9)); however, taller-than-wide was a feature reported in only 12 of the included studies. Other ultrasound findings that were significant in differentiating benign from malignant nodules were infiltrative margins, internal calcification, hypoechoic features, solid features, and increased blood flow. Conversely, the sonographic features with the highest diagnostic odds ratio of a benign nodule were characterized by a spongy appearance (12 (95% confidence interval 0.61–234.3)), cystic features, and isoechogenicity. In a meta-analysis by Campanella et al., nodule size ≥ 4 cm and single nodules were cited as significant risk factors for thyroid cancer diagnosis, in addition to the sonographic features in JSUM ultrasound criteria and the ultrasound findings reported by Brito et al. In addition, the meta-analysis by Remonti et al. included the presence of elasticity as an additional risk factor. The common malignant ultrasound findings in these reports were a higher-than-wide shape, microcalcification, irregular borders, internal hypoechogenicity, solid nodules, and internal blood flow.

### 2.3. Risk Assessment of Microcarcinoma Basd on Ultrasound Findings

Recently, surgical treatment for thyroid cancer has become more conservative. In Japan, for microcarcinomas of 10 mm or less, active surveillance has additionally been performed in cases without metastasis, tumor growth, or invasion of the recurrent nerve or trachea [11]. Fukuoka et al. investigated risk factors for tumor growth through multivariate analysis and found that stronger calcification patterns, which were defined as macroscopic (coarse) calcification (>1 mm), agglutinated calcification with acoustic shadow, and rim calcification completely aligned along the rim of the tumors, as well as poorer vascularity, were correlated significantly with non-progressive disease [12]. Sugitani et al. performed a multivariate analysis of active surveillance in microcarcinoma over a mean period of 5 years and found that extrathyroidal invasion, lymph node metastasis, and poorly differentiated composition were significantly related to adverse outcomes, such as disease-free survival [13].

As clinical lymph node metastasis and extrathyroid extension are important risk factors for poor prognosis, even in patients with papillary thyroid microcarcinoma, the accurate evaluation of these factors is recommended [4]. Careful sonographic evaluation is recommended to detect lymph node metastasis and extrathyroid extension; however, ultrasonography is not sufficient to detect central metastatic lymph nodes. Ito et al. showed that in surgically treated patients with micropapillary carcinoma, ill-defined tumor edges and fine strong echoes on ultrasound images were associated with thyroid cancer recurrence and lateral lymph node metastasis [14]. Recent studies of micropapillary thyroid carcinomas showed that the risk factors of lymph node metastasis in microcarcinoma included male sex, younger age (<45 years), larger tumor size, microcalcification, extrathyroidal invasion, hypoechogenicity within the tumor, and multicentricity [15,16,17,18,19,20,21]. In addition, Gur et al. reported that multiple micropapillary carcinomas had significantly larger tumor diameters and greater invasion of the surgical margins [22]. These reports suggest that ultrasound findings should be carefully examined when deciding on the diagnosis and treatment plan for thyroid cancer.

## 3. Implementation Criteria for FNAC

### 3.1. Criteria in Japan

It is known that thyroid cancer is frequently detected during screening via ultrasonography and that latent cancer is detected extremely frequently during autopsy. Therefore, the JABTS published criteria for the management of thyroid nodules and presented a diagnostic flowchart for nodular thyroid disease to provide a standard [23,24]. This flowchart categorizes nodular lesions into solid and cystic nodules, providing the criteria for each of them. These criteria are also followed in the guidelines for thyroid diseases issued by the Japanese Thyroid Association. In 2016, the JABTS revised the guidebook (3rd edition), and only the criteria for the management of cystic nodules were revised [25].

According to this criterion, FNAC is recommended for nodules measuring 5.1–10.0 mm in diameter that are strongly suspicious for thyroid carcinoma according to the JSUM diagnostic criteria, that is, when most of the malignant sonographic findings are observed (Figure 1) [24,25]. Typical pathological diagnoses of malignant nodules diagnosed from this category may include the diffuse sclerosing variant of PTC (DSV-PTC) or PTC with extrathyroidal extension or lymph node metastasis (Figure 2). FNAC was also recommended for nodules sized 10.1–20.0 mm in diameter that were suspicious for carcinoma according to the above criteria, i.e., when some of the malignant sonographic findings were observed (Figure 1). Typical pathological diagnoses of malignant nodules in this category may include PTC, the follicular variant of PTC (FV-PTC), or FTC (Figure 2). All nodules > 20 mm in diameter and nodules showing spongiform patterns were subjected to FNAC (Figure 1). Typical pathological diagnoses of nodules diagnosed from this category may include the cribriform-morular variant of PTC (CMV-PTC), follicular adenoma, adenomatous nodules, or cysts (Figure 2). Although nodules measuring < 5.1 mm in diameter were not recommended for FNAC, cases with obvious cervical lymph node metastases, family histories of genetic MTC, histories of treatment for high-risk thyroid cancer, or any symptoms indicating cervical extension of thyroid cancer were recommended for FNAC.

If cystic lesions are observed, they are classified according to the presence or absence of a solid portion. If there is no solid component, FNAC is not recommended for ≤20.0 mm, and aspiration of the cyst fluid is considered for cysts of 20.1 mm or greater in diameter to relieve symptoms (Figure 3) [25]. The ratio of the solid component in the maximum cross-section is divided into 50% or more and less than 50%, and the former follows the above-mentioned management criterion for solid nodules. In the latter case, FNAC is not recommended when the maximum diameter, including the cystic component, is 5 mm or less. For nodules with a maximum diameter of ≥5.1 mm and ≤20.0 mm, FNAC is recommended when a nodule is suspected to have extramural infiltration, even if the solid component is less than 5.1 mm. FNAC is recommended when there are multiple malignant findings among the irregular surfaces of solid components, fine multiple high-echo spots, and increased blood flow for a nodule with 5.1 to 10.0 mm in diameter of a solid component. For a nodule with a solid area exceeding 10.0 mm, FNAC is recommended when any of the above malignant findings are present. If the maximum diameter of a cystic nodule is 20.1 mm or more, in principle, FNAC is performed.

### 3.2. Criteria in Other Countries

Guidelines for the implementation of FNAC have been published by the American Thyroid Association (ATA) [26], National Comprehensive Cancer Network (NCCA) [27], and American College of Radiology (ACR) [28] from the United States, and the European Society for Medical Oncology (ESMO) [29] and the European Thyroid Association (ETA) [30] from Europe. All of these guidelines use US findings to classify risk according to the size and sonographic features of thyroid nodules (Table 2).

In the ATA guidelines [26], FNAC is recommended for thyroid nodules ≥10 mm with a highly suspicion pattern, that is, solid hypoechoic nodules or solid hypoechoic components of a partially cystic nodule with one or more of the following features: irregular margins (infiltrative, microlobulated), microcalcifications, a taller-than-wide shape, rim calcifications with small extrusive soft tissue component, evidence of extrathyroidal extension, and with an intermediate suspicion pattern, that is, hypoechoic solid nodule without any malignant findings. In the case of a low-suspicion pattern, that is, an isoechoic or hyperechoic solid nodule, or partially cystic nodule with eccentric solid areas without any malignant findings, FNAC is recommended for nodules ≥ 15 mm in diameter. In the case of a very-low-suspicion pattern, that is, spongiform or partially cystic nodules without any malignant findings, the performance of FNAC is considered. Although FNA is not recommended in principle for nodules measuring 10 mm or less, these guidelines mention that FNA can be considered at lower size-cutoffs for all of the sonographic appearances described above.

The ACR published the ACR TI-RADS system, which provides guidance regarding the management of thyroid nodules [28]. The ultrasound features in the ACR TI-RADS, which include composition, echogenicity, shape, margin, and echogenic foci, are categorized as benign, minimally suspicious, moderately suspicious, or highly suspicious for malignancy. The total points determine the nodule’s ACR TI-RADS level, which ranges from TR1 (benign) to TR5 (high suspicion of malignancy). Recommendations for FNAC or ultrasound follow-up are based on a nodule’s ACR TI-RADS level and its maximum diameter. FNAC is recommended for ≥10 mm nodules for risk levels TR5 (highly suspicious), ≥15 mm for TR4 (moderately suspicious), and ≥25 mm for TR3 (mildly suspicious). However, FNAC is not recommended for TR2 (not suspicious) or TR1 (benign).

### 3.3. Comparison of FNAC Guidelines with Case Studies

To compare the guidelines of Japan and USA in the FNAC implementation criteria, we collected case studies with actual sonographic images (shows as cases A to D–Figure 4). Table 3 shows the evaluations of sonographic characteristics and the estimated risk of malignancy based on the guidelines of the JABTS, ATA, and ACR. In addition, the recommended strategy, FNAC, or follow-up was assessed using the guidelines of the JABTS, ATA, and ACR for each assumed nodule size (Table 3). In Case A, evaluations with the JABTS, ATA, and ACR guidelines were strongly suspected to be suspicious of malignancy. However, the JABTS guideline recommends FNAC, that of ATA considers FNAC, and ACR does not recommend FNAC if the nodule size is ≤10 mm. Since cases similar to Case A are not subject to active surveillance even if they are categorized as micropapillary thyroid carcinoma, FNAC is recommended for such cases. Case B with hypoechogenicity in the nodule was estimated to have an intermediate risk in all guidelines, and FNAC was recommended if the nodule size was greater than 10 mm in the JABTS and ATA guidelines and >15 mm in ACR. In Case C, in which the thyroid nodule exhibits a spongiform pattern, the guidelines of ATA and JABTS suspect the risk to be very low and not suspicious for malignancy, respectively, and do not recommend FNAC up to 20 mm in diameter, whereas the ACR guidelines estimate such cases to be moderately suspicious and recommend FNAC when the nodule is >15 mm. Case D showed no malignant findings on sonographic characteristics, and the guidelines of JABTS and ACR do not recommend FNAC up to 20 mm in diameter, but the ATA guideline estimate this as having low suspicion and FNAC is recommended for nodules greater than 15 mm. Thus, although there are slight differences in the recommendations for FNAC among the guidelines of the JABTS, ATA, and ACR, similar management for nodules is recommended.

## 4. Conclusions

Thyroid cancer is a malignant tumor with a risk of overdiagnosis in the case of very low-risk cancers. In the diagnosis of thyroid cancer, detailed ultrasonography should be performed to assess the risk of tumor size and ultrasound findings before determining the indication for FNAC. Although the ultrasonographic findings in B-mode imaging for the diagnosis of thyroid nodules are almost the same in each country, some reports show that intranodular blood flow and elastography are also useful [8,9,10]. Therefore, further elucidation is required in order to evaluate other criteria with intranodular blood flow or elastography. The FNAC implementation criteria are considered to be similar for each criterion, even though there are some minor differences. Detailed and accurate evaluation of the ultrasound findings of the nodule, rather than classification based on size alone, is considered necessary for the management of thyroid nodules based on the FNAC implementation criteria.

In Japan, JABTS launched the Board-Certified Thyroidologist for Ultrasound-Guided Diagnostic Biopsy initiative for medical doctors and the Board-Certified Coordinator for Thyroid Ultrasound-Guided Diagnostic Biopsy initiative for medical staff. These efforts will enhance the standardization of thyroid nodule management in Japan.

## Figures and Tables

**Figure 1 cancers-13-04629-f001:**
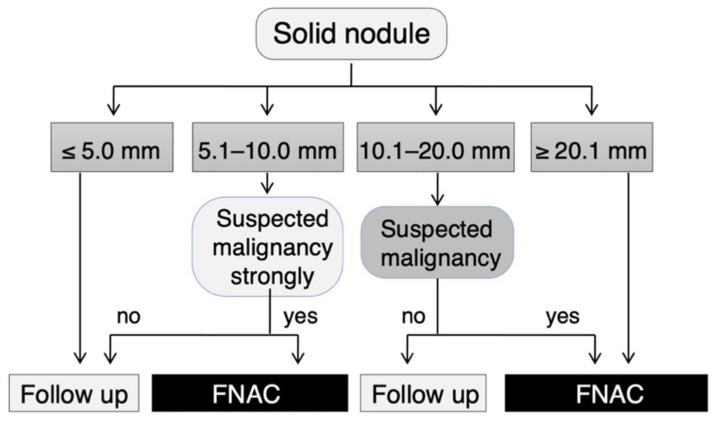
Flow chart for the management of solid thyroid nodules in Japan. This flow chart, showing the management of solid thyroid nodules, was published by the Japan Association of Breast and Thyroid Sonology. For nodules with a spongiform pattern, FNAC is not recommended up to 20.0 mm, with or without malignant findings. Although nodules measuring < 5.1 mm in diameter were not recommended for FNAC, cases with obvious cervical lymph node metastases, family histories of genetic MTC, histories of treatment for high-risk thyroid cancer, or any symptoms indicating cervical extension of thyroid cancer were recommended for FNAC. This figure was created based on Figure 5 of Chapter V in *Thyroid Ultrasound–A Guidebook for Diagnosis and Management*, 3rd edition [25], with the permission of Nankodo Corp.

**Figure 2 cancers-13-04629-f002:**
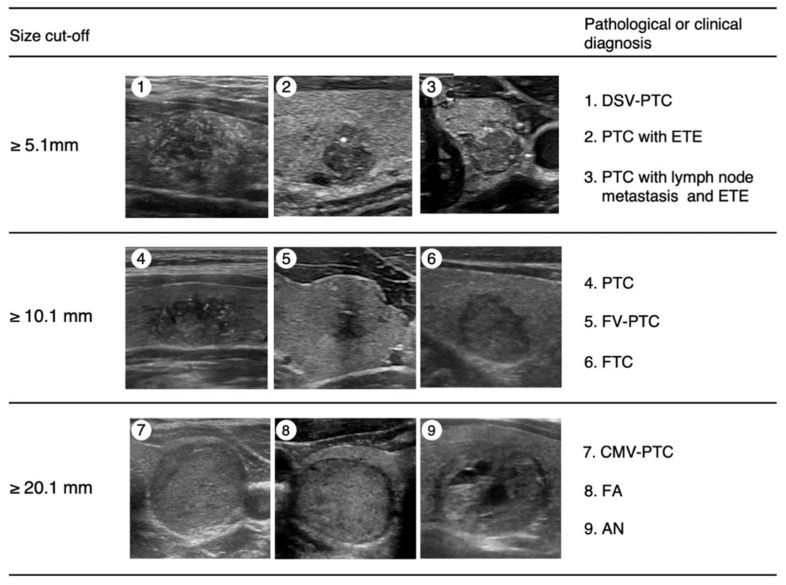
Cut-off sizes of nodules for FNAC and corresponding representative ultrasound images. Images (1–3): Representative ultrasound images of nodules for which FNAC is recommended at a diameter of 5.1 mm or greater. All images exhibited an irregular shape, ill-defined and jagged border, low and heterogeneous internal echo level, and fine strong echoes. Images (4–6): Representative ultrasound images of nodules for which FNAC is recommended at a diameter of 10.1 mm or greater. Image (4) shows a jagged border, low internal echo level, and multiple fine strong echoes; image (6) shows an ill-defined border and fine and coarse strong echoes; image (7) shows a jagged border and irregular marginal hypoechoic zone. Images (7–9): Representative ultrasound images of nodules for which FNAC is recommended at a diameter of 20.1 mm or greater. Images (7,8) show no malignant findings; image (8) shows a spongiform pattern and ill-defined border. Pathological or clinical diagnoses in each image are shown in the right column. PTC, papillary thyroid carcinoma; DSV-PTC, cribriform-morular variant of PTC; ETE, extrathyroidal extension; FV-PTC, follicular variant of PTC; FTC, follicular carcinoma; CMV-PTC, cribriform-morular variant of PTC; FA, follicular adenoma; AN, adenomatous nodule.

**Figure 3 cancers-13-04629-f003:**
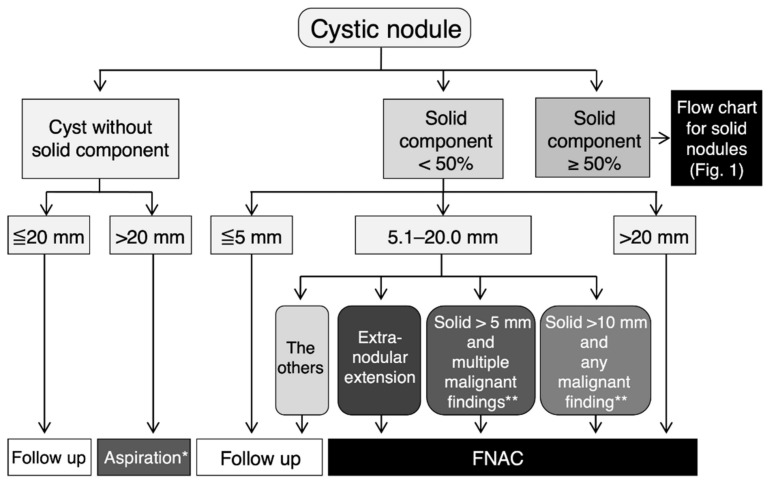
Flow chart for the management of cystic thyroid nodules in Japan. This flow chart for the management of cystic thyroid nodules was published by the Japan Association of Breast and Thyroid Sonology. This figure was created based on Figure 4 of Chapter V in Thyroid Ultrasound–A Guidebook for Diagnosis and Management, 3rd edition [25], with the permission of Nankodo Corp. * Aspiration of cyst fluid is considered for cysts that are 20.1 mm or greater in diameter to relieve symptoms. ** Irregular surface of solid component, multiple fine high-echo spots, and increased blood flow.

**Figure 4 cancers-13-04629-f004:**
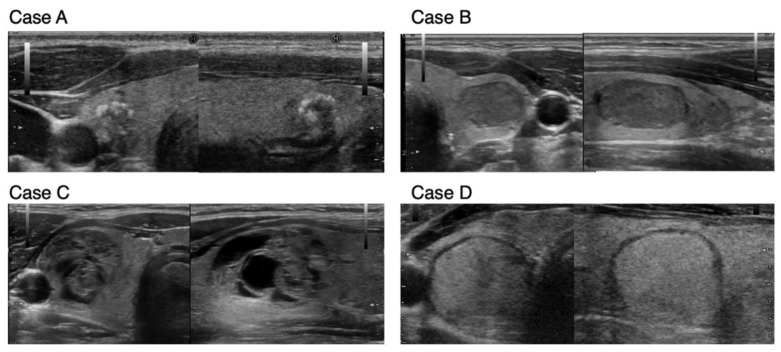
Sonographic images of **Case** (**A**–**D**) for case studies in Table 3. The left and right panels of each case show the transverse and longitudinal images, respectively. The taller-than-wide sign was evaluated using transverse images.

**Table 2 cancers-13-04629-t002:** Guidelines for the implementation of FNAC.

Publisher	5–10 mm	10–15 mm	15–20 mm	>20 mm
JABTS (2016) [25]	Strongly suspicious (most of the malignant sonographic findings were observed)	Suspicious (some of the malignant sonographic findings were observed)		Not suspicious (Nodule without malignant findings or spongiform nodule with any malignant findings)
ATA (2016) [26]	Sonographically suspicious subcentimeter thyroid nodule without evidence of extrathyroidal extension or sonographically suspicious lymph nodes may be observed. FNA can be considered at lower size cutoffs for all of the sonographic appearances	Highly suspicion pattern (Solid hypoechoic nodule or solid hypoechoic component of a partially cystic nodule with one or more of the following features) Intermediate suspicious pattern(Hypoechoic solid nodule without any malignant findings)	Low suspicion pattern (Isoechoic or hyperechoic solid nodule, or partially cystic nodule with eccentric solid areas without any malignant findings)	Very low suspicion pattern (Spongiform or partially cystic nodules without any malignant findings)
NCCN (2019) [27]	Not recommended	Solid nodule with suspicious US features Mixed cystic-solid nodule with suspicious US features if solid component is >1 cm	Solid nodule without suspicious US features Mixed cystic-solid nodule without suspicious US features if solid component is >1.5 cm	Spongiform nodule
ACR (2017) [28]	Not recommended	Highly suspicious (≥7/14 points)	(15–25 mm) Moderately suspicious (4–6/14 points)	(≥25 mm) Mildly suspicious (3/14 points)
ESMO (2019) [29]	Preoperative FNA for cytology is not required for nodules measuring ≤ 1 cm	Same as ATA	Same as ATA	Same as ATA
ETA (2017) [30]	Patients with < 1 cm nodules with highly suspicious US features and no abnormal lymph nodes can have the choice of FNA or observation. In case of proven growth or detection of a suspicious lymph node, FNA should be performed.	Nodule with high-risk US features	Nodule with intermediate-risk US features	Nodule with low-risk US features

**Table 3 cancers-13-04629-t003:** Comparison of FNAC guidelines with case studies. Malignant ultrasound findings and FNAC recommendations are presented in bold.

Case		A		B		C		D	
Shape		**Irregular** **Taller than Wide**		Regular Wider than Tall		**Irregular** **Taller than Wide**		Regular Wider than Tall	
Border	Definition	**Ill-defined**		Well-defined		**Ill-defined**		Well-defined	
	Character	**Jagged**		Smooth		**Jagged**		Smooth	
Internal echoes	Level	**Hypoechoic**		**Hypoechoic**		Isoechoic		Isoechoic	
	Heterogeneity	**Heterogeneous**		**Heterogeneous**		Homogeneous		Homogeneous	
Strong echoes		**Multiple and fine**		None		None		None	
Marginal hypoechoic zone (halo)		**Absent**		**None**		**None**		Regular	
Solid/cystic		Solid		Solid		Spongiform		Solid	
Estimated risk of malignancy	JABTS	Strongly suspicious		Suspicious		Not suspicious		Not suspicious	
	ATA	High suspicion		Intermediate suspicion		Very low suspicion		Low suspicion	
	ACR	Highly suspicious		Moderately suspicious		Moderately suspicious		Mildly suspicious	
Assumed size		9 mm	11 mm	14 mm	16 mm	19 mm	26 mm	19 mm	26 mm
Recommended strategy	JABTS	**FNAC**	**FNAC**	**FNAC**	**FNAC**	Follow up	**FNAC**	Follow up	**FNAC**
	ATA	**Consider FNAC**	**FNAC**	**FNAC**	**FNAC**	Follow up	**FNAC**	**FNAC**	**FNAC**
	ACR	Follow up	**FNAC**	Follow up	**FNAC**	**FNAC**	**FNAC**	Follow up	**FNAC**

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
