# Peer review of "Diagnostic Strategies for Thyroid Nodules Based on Ultrasonographic Findings in Japan"

_cancers, 2021, doi:10.3390/cancers13184629_

Round 1
Reviewer 1 Report
The article is interesting and well written and I think can be accepted in the present form
Author Response
We wish to express our appreciation to the reviewer for his or her kind review.
Reviewer 2 Report
The charts and algorithyms are very interesting and could be very practical to use. Radiological imagings are highly qualified.
Author Response

(The authors gave the same response as above.)

Reviewer 3 Report
This is a well written article on the ultrasonographic characteristics of thyroid cancer in Japan. It is well structured. I do have several comments:
-line 126: pleas elaborate what stronger calcifications pattern means (macro-, and rim calcifications) as not clear for readers not familiar with that study
-line 127: that sentence regarding reference 14 should be deleted and rephrased. Ito et al were recommending surgery for all PMCs touching the trachea or located in the course of the recurrent laryngeal nerve. However in this article, the find that 7mm is a cutoff and they report observation reasonable for low risk PMCs for touching trachea or invading RN based on imaging studies. The way the authors use high risk in the text, and the way they have phrased their conclusion do not convey the same message as the authors of reference 14.
-line 130:extrathyroidal extension, LARGE LN metastasis>2 cm and poorly differentiated component were not risk factors for subsequent surgery, but for adverse outcomes, specifically disease-free survival. Authors should correct
-figure 2:there is an error in the label for images 4-6
-line 168: the authors should remove the cystic nodules since they are talking about cystic nodules later in the manuscript. Otherwise, they should also reference figure 3 where they talk about cystic nodules
-line 199: do the authors mean simple aspiration instead of FNAC? If so, they should correct. If not, they should elaborate on how they will have cytology and what is the yield.
-Figure 3: Also, on the same topic, why are the authors proposing a 2cm cutoff? I suppose it is not for cancer risk (as they are cysts without any solid component). If the patient is symptomatic from a large cyst, then it should be drained, but I am not sure where the 2 cm cutoff is coming from. I do not have access to Reference 26 which I suppose is the source for that statement. However, that statement should be modified to something that is closer to clinical practice
-line 218: would remove the word recently as the ATA guidelines have been out for 5 years.
-line 251: it's Japan and USA, as both ATA and ACR are US organizations
-lines 254, 256, 273: ACR, not ACS.
-table 3: I like how this is presented. The authors should maybe consider dividing the cases with lines so that's clear to the reader that the two different sizes are about the same case.
Author Response
Response to Reviewer #3
We wish to express our appreciation to the reviewer for his or her insightful comments that have helped us to improve the manuscript significantly.
Comment 1:
-line 126: pleas elaborate what stronger calcifications pattern means (macro-, and rim calcifications) as not clear for readers not familiar with that study
Response for Comment 1:
We thank the reviewer for this pertinent comment. We added the definition of stronger calcification in this sentence.
Lines 125-129
Fukuoka et al. investigated risk factors for tumor growth by multivariate analysis and found that stronger calcification patterns, which were defined as macroscopic (coarse) calcification (>1 mm), agglutinated calcification with acoustic shadow, and rim calcification completely aligned along the rim of the tumors, and poorer vascularity correlated significantly with non-progressive disease 13.
Comment 2:
-line 127: that sentence regarding reference 14 should be deleted and rephrased. Ito et al were recommending surgery for all PMCs touching the trachea or located in the course of the recurrent laryngeal nerve. However in this article, the find that 7mm is a cutoff and they report observation reasonable for low risk PMCs for touching trachea or invading RN based on imaging studies. The way the authors use high risk in the text, and the way they have phrased their conclusion do not convey the same message as the authors of reference 14.
Response for Comment 2:
We agree with the reviewer's comment. We deleted this sentence from this manuscript.
Comment 3:
-line 130:extrathyroidal extension, LARGE LN metastasis>2 cm and poorly differentiated component were not risk factors for subsequent surgery, but for adverse outcomes, specifically disease-free survival. Authors should correct
Response for Comment 3:
We appreciate the reviewer for the appropriate comment.
According to the reviewer’s comment, we have revised the sentences the reviewer pointed out.
Lines 130-133
Sugitani et al. performed multivariate analysis of active surveillance in microcarcinoma over a mean period of 5 years and found that extrathyroidal invasion, lymph node metastasis, and poorly differentiated composition were significantly related to adverse outcomes, such as disease-free survival [13].
Comment 4:
-figure 2:there is an error in the label for images 4-6
Response for Comment 4:
There were a few errors where spaces were mistakenly not included between sentences, and these have been corrected.
Comment 5:
-line 168: the authors should remove the cystic nodules since they are talking about cystic nodules later in the manuscript. Otherwise, they should also reference figure 3 where they talk about cystic nodules
Response for Comment 5:
We appreciate the reviewer for the appropriate comment.
According to the reviewer’s comment, we deleted this part of the sentence from this manuscript.
Comment 6:
-line 199: do the authors mean simple aspiration instead of FNAC? If so, they should correct. If not, they should elaborate on how they will have cytology and what is the yield.
Comment 7:
-Figure 3: Also, on the same topic, why are the authors proposing a 2cm cutoff? I suppose it is not for cancer risk (as they are cysts without any solid component). If the patient is symptomatic from a large cyst, then it should be drained, but I am not sure where the 2 cm cutoff is coming from. I do not have access to Reference 26 which I suppose is the source for that statement. However, that statement should be modified to something that is closer to clinical practice
Response for Comment 6 and 7:
We appreciate the reviewer for the comments.
According to the reviewer’s comment, we revised the sentence and the legend for Fig. 3 as follows.
Lines 197-199
If there is no solid component, FNAC is not recommended for ≤ 20.0 mm, and aspiration of cyst fluid is considered for cysts 20.1 mm or greater in diameter to relieve symptoms. (Fig. 3) [25].
Legend for Figure 3 in lines 214 to 215
* Aspiration of cyst fluid is considered for cysts 20.1 mm or greater in diameter to relieve symptoms.
Comment 8:
-line 218: would remove the word recently as the ATA guidelines have been out for 5 years.
Response for Comment 8:
We appreciate the reviewer for the appropriate comment.
According to the reviewer’s comment, we removed the word “recently” form this sentence.
Comment 9 and 10:
-line 251: it's Japan and USA, as both ATA and ACR are US organizations
-lines 254, 256, 273: ACR, not ACS.
Response for Comment 9 and 10:
We appreciate the reviewer for the comments.
According to the reviewer’s comment, we corrected these sentences.
Comment 11:
-table 3: I like how this is presented. The authors should maybe consider dividing the cases with lines so that's clear to the reader that the two different sizes are about the same case.
Response for Comment 11:
We appreciate the reviewer for the helpful comments.
According to the reviewer’s comment, we added lines between cases A to D in Table 3.
Reviewer 4 Report
Shimura et al. article outlines the diagnostic criteria and management guidelines in Japan in comparison with those published in other countries. The advantage of the article are very transparent tables on what the diagnostic procedure should be. The paper is clear and interesting and the discussion is relevant. The paper should be published in Cancers after a minor revision.
Minor remarks:
Lines 55-60: reference #5 is not relevant to the text – please omit the reference
Lines 75-76: “These criteria were first published in 1991, were revised in 1999 and have been 75 revised again.” Please cite all three publications.
Lines 85-87: Please cite the source of the current diagnostic criteria.
Lines 87: delete “papillary carcinoma” because it is listed twice in the same sentence
Table 1: left column, 3rd line: “border”
Line 125: delete “in Japan” because it is listed twice in the same sentence
Line 143: instead of “multiplicity” use “multicentricity”
Table 3: The 2nd column is not aligned
Table 3: The 2nd column: instead of “Heterogeity” use “Heterogeneity”
Lines 281-283: “Although the ultrasonographic findings for the diagnosis of thyroid nodules are almost the same in each country, some reports show that intranodular blood flow and elastography are also useful. Therefore, further elucidation is required.” Please rewritte. You are describing criteria and guidelines of different professional societies. List the common criteria. Further studies are needed in order to evaluate other criteria as intranodular blood flow or elastography.
Author Response
We wish to express our appreciation to the reviewer for his or her insightful comments that have helped us to improve the manuscript significantly.
Comment 1:
Lines 55-60: reference #5 is not relevant to the text – please omit the reference
Response for Comment 1:
We agree with the reviewer's comment. We deleted this reference.
Comment 2:
Lines 75-76: “These criteria were first published in 1991, were revised in 1999 and have been 75 revised again.” Please cite all three publications.
Response for Comment 2:
The references for the criteria published in 1991 and 1999 mentioned in lines 75-76 were written only in Japanese and is not considered to be an important for this chapter. Therefore, we deleted this sentence.
Comment 3 and 4:
Lines 85-87: Please cite the source of the current diagnostic criteria.
Lines 87: delete “papillary carcinoma” because it is listed twice in the same sentence
Response for Comment 3 and 4:
We appreciate the reviewer for pointing this out. We added Ref #5 as the reference and corrected this error.
Comment 5 to 10:
Table 1: left column, 3rd line: “border”
Line 125: delete “in Japan” because it is listed twice in the same sentence
Line 143: instead of “multiplicity” use “multicentricity”
Table 3: The 2nd column is not aligned
Table 3: The 2nd column: instead of “Heterogeity” use “Heterogeneity”
Response for Comment 5 to 10:
We appreciate the reviewer for pointing this out. We corrected these errors.
Comment 11:
Lines 281-283: “Although the ultrasonographic findings for the diagnosis of thyroid nodules are almost the same in each country, some reports show that intranodular blood flow and elastography are also useful. Therefore, further elucidation is required.” Please rewritte. You are describing criteria and guidelines of different professional societies. List the common criteria. Further studies are needed in order to evaluate other criteria as intranodular blood flow or elastography.
Response for Comment 11
We appreciate the reviewer for the appropriate comment. We rewrote these sentences as follows.
Lines 281-283
Although the ultrasonographic findings in B-mode imaging for the diagnosis of thyroid nodules are almost the same in each country, some reports show that intranodular blood flow and elastography are also useful [8-10]. Therefore, further elucidation is required in order to evaluate other criteria with intranodular blood flow or elastography.